# Initial Steps towards Spatiotemporal Signaling through Biomaterials Using Click-to-Release Chemistry

**DOI:** 10.3390/pharmaceutics14101991

**Published:** 2022-09-21

**Authors:** Merel Gansevoort, Jona Merx, Elly M. M. Versteeg, Isidora Vuckovic, Thomas J. Boltje, Toin H. van Kuppevelt, Willeke F. Daamen

**Affiliations:** 1Department of Biochemistry, Radboud Institute for Molecular Life Sciences (RIMLS), Radboud University Medical Center, Geert Grooteplein 28, 6525 GA Nijmegen, The Netherlands; 2Institute for Molecules and Materials, Synthetic Organic Chemistry, Radboud University, 6500 GL Nijmegen, The Netherlands

**Keywords:** click chemistry, bio-orthogonal chemistry, trans-cyclooctene, tetrazine, controlled release, collagen scaffolds, wound healing, regenerative medicine

## Abstract

The process of wound healing is a tightly controlled cascade of events, where severe skin wounds are resolved via scar tissue. This fibrotic response may be diminished by applying anti-fibrotic factors to the wound, thereby stimulating regeneration over scarring. The development of tunable biomaterials that enable spatiotemporal control over the release of anti-fibrotics would greatly benefit wound healing. Herein, harnessing the power of click-to-release chemistry for regenerative medicine, we demonstrate the feasibility of such an approach. For this purpose, one side of a bis-N-hydroxysuccinimide-trans-cyclooctene (TCO) linker was functionalized with human epidermal growth factor (hEGF), an important regulator during wound healing, whereas on the other side a carrier protein was conjugated—either type I collagen scaffolds or bovine serum albumin (BSA). Mass spectrometry demonstrated the coupling of hEGF–TCO and indicated a release following exposure to dimethyl-tetrazine. Type I collagen scaffolds could be functionalized with the hEGF–TCO complex as demonstrated by immunofluorescence staining and Western blotting. The hEGF–TCO complex was also successfully ligated to BSA and the partial release of hEGF upon dimethyl-tetrazine exposure was observed through Western blotting. This work establishes the potential of click-to-release chemistry for the development of pro-regenerative biomaterials.

## 1. Introduction

In adult humans, wounds are often resolved with a scar. While scarring is a normal physiological response, severe forms of scarring (fibrosis) negatively impact the patients’ quality of life [1,2,3]. Nearly all processes in the body are regulated by signaling factors, such as growth factors, cytokines, hormones, neurotransmitters, etc. From fetal development to wound healing in adults, signaling factors orchestrate biological processes and deviations from the norm can have serious consequences [4,5,6,7,8]. To illustrate, skin wound healing consists of phases orchestrated by signaling factors, e.g., hemostasis, inflammation, cellular proliferation, and matrix deposition and remodeling, eventually resulting in scar tissue [9,10,11,12]. Skin wounds where both the dermis and epidermis are damaged, e.g., large third-degree burn wounds, regularly result in contractures and fibrosis [13,14,15,16].

Fibrosis is largely caused by the overactivation of myofibroblasts [17]. These contractile cells are important players in wound healing, contracting the wound area to accelerate wound closure and depositing new extracellular matrix components to reconstruct lost tissue. As with all cells, the activation of myofibroblasts is regulated by signaling factors, most notably by transforming growth factor β (TGFβ) [18,19]. To combat fibrosis, targeting myofibroblast activation with effector molecules has demonstrated potential. For example, the application of fibroblast growth factor 2 (FGF2)—an inhibitor of the TGFβ1-induced fibroblast-to-myofibroblast transition—to second-degree burn wounds in humans increased wound-healing speed and reduced the severity of scarring [20,21]. Previously, type I collagen scaffolds functionalized with heparin, a prominent glycosaminoglycan; fibroblast growth factor 2 (FGF2); and vascular endothelial growth factor (VEGF) showed beneficial effects towards full-thickness skin wound healing in rats and sheep [22,23]. Epidermal growth factor (EGF), a regulator of keratinocyte proliferation and migration, was one of the first growth factors to be applied in wound healing and it continues to be an attractive therapeutic agent [24]. EGF conjugated to hyaluronic acid led to increased transdermal transport of EGF in the skin wounds of rats and improved wound healing [25]. Improved skin wound healing in rats was also observed when EGF was incorporated into a nanodrug delivery system using curcumin-loaded chitosan nanoparticles [26].

The potential of signaling factors to guide cells and steer biological processes makes them prime candidates for use in skin wound healing and regeneration [27,28,29]. In particular, growth factors are very effective at steering cell behavior. However, their application in wound healing has been limited in part due to challenges relating to growth factor delivery methods [30]. Topically applied growth factors have a short half-life that limits their window of activity [31]. Similarly, injection is a straight forward delivery method but growth factors can easily diffuse away from the injection site, additionally resulting in limited bioactivity [32]. These methods inevitably lead to the requirement of higher dosages in order to ensure that effects are sustained over a longer period of time, which is accompanied with negative side effects [33]. The local and sustained application of growth factors through the use of functionalized biomaterials provides an improved delivery method [34,35,36]. Naturally derived biomaterials such as alginate, agarose, and gelatin have been used to develop drug delivery systems with sustained drug release and improved drug half-lives [37,38,39]. Drug release kinetics can be controlled by varying parameters such as the drug loading, drug hydrophobicity, biomaterial concentration, biomaterial crosslinking, and the incorporation of additives. Controlled release may also be dependent on environmental stimuli such as pH, heat, light, or electrical current [40,41,42]. In wounded skin, signaling molecules should be released at a specific time and place to guide the tissue towards regeneration and away from fibrosis. Thus, the next step in the improvement of functionalized biomaterials should focus on the protection and absolute control over the time and place of the release of signaling factors [43]. The field of click chemistry, and particularly “click-to-release” chemistry, offers unique opportunities for developing such spatiotemporal signaling scaffolds [44].

Various types of click chemistry exist; of these types, inverse electron demand Diels-Alder (IEDDA) reactions are a type of bio-orthogonal reaction uniquely suitable for use in vivo. These highly selective ‘click’ reactions between 1,2,4,5-tetrazines and dienophiles meet the requirements for in vivo uses due to their fast kinetics, high selectivity, catalyst-free reaction kinetics, and lack of cytotoxicity [45]. An intriguing avenue for the on-demand release of bioactive compounds was started with the development of dissociative bio-orthogonal reactions [46,47]. Such ‘click-to-release’ reactions have been successfully used to develop novel antibody–drug conjugates, which enable selective in vivo targeting through antibody recognition followed by controlled release upon the introduction of a chemical trigger [48]. The system in question used a trans-cyclooctene (TCO), which was coupled to the cytostatic drug doxorubicin via a carbamate located on the allylic position (the ‘leaving group’), where doxorubicin release was triggered upon tetrazine ligation [49,50].

Based on this approach, we propose a customizable platform based on bio-orthogonal ‘click-to-release’ chemistry that enables the spatiotemporal functionalization of biomaterials with growth factors and small molecules. By designing a ‘click-to-release’ linker where the leaving group is a signaling molecule that is interchangeable depending on the need of the regenerating tissue, a new category of customizable and off-the-shelf biomaterials may be developed.

To this end, we utilized a bis-N-hydroxysuccinimide (NHS)-functionalized TCO molecule (TCO, Figure 1) [49]. The hypothesis is that a primary amine group on the bioactive compound will covalently attach to the TCO via the NHS group at the allylic carbonate, whereas type I collagen is bound analogously via the remaining NHS group, which is less reactive due to the steric hindrance provided by the methyl group. Upon the addition of a tetrazine, the click reaction with TCO initiates an electron cascade, resulting in the release of the bioactive compound. Primary amine groups may be present on the growth factors and small molecules, ensuring that a variety of bioactive compounds may be used in this approach. Here, we offer the proof of the feasibility of this novel approach by functionalizing porous type I collagen scaffolds with human epidermal growth factor (hEGF).

## 2. Materials and Methods

### 2.1. Preparation of Type I Collagen Scaffolds

A suspension of 0.8% (*w*/*v*) insoluble type I collagen fibrils from bovine tendon in 0.25 M acetic acid (Biosolve BV, Valkenswaard, The Netherlands) was homogenized and air bubbles were removed under vacuum. The collagen suspension was then frozen at −20 °C in 50 mL polystyrene tubes and lyophilized, resulting in porous collagen scaffolds [51].

### 2.2. Reaction Components and Conditions

The bis-NHS-TCO linker (TCO), with a theoretical molecular weight of 422.39 g/mol (C_19_H_22_N_2_O_9_), was provided by Tagworks Pharmaceuticals (Nijmegen, The Netherlands) and stored at −20 °C. TCO was dissolved to 10 mg/mL in dimethyl sulfoxide (DMSO, Merck, Darmstadt, Germany) and aliquots were kept at −20 °C for a maximum of 7 days due to the instable nature of the compound. The 3,6-dimethyl-1,2,4,5-tetraazine (tetrazine) was purchased from Merck and stored according to the manufacturer’s instructions at a concentration of 0.1 M in DMSO. Recombinant human EGF (hEGF) was purchased from Peprotech (Cranbury, NJ, USA) and dissolved to 1 mg/mL in H_2_O according to the manufacturer’s instructions. Bovine serum albumin fraction V (BSA) was obtained from Merck. All reactions were performed at room temperature and protected from light in 1.5 mL Protein LoBind Eppendorf tubes (Eppendorf AG, Hamburg, Germany) under constant agitation using a platform attached to a Vortex-Genie^®^ 2, set to shake at speed of 2 (Scientific Industries, Bohemia, NY, USA). Incubation times ranged from 4 to 16 h and are specified for each experiment. The components were combined using the following molar ratios: 1 mole of hEGF to 50 moles of TCO and 1 mole of TCO to 10 moles of tetrazine (1:50:500).

### 2.3. Ligation of hEGF to TCO and Release of hEGF following Tetrazine Exposure

Reaction products of hEGF, TCO, and tetrazine were detected using mass spectrometry. To this end, 100 μM hEGF was ligated overnight (16 h) to 5 mM TCO followed by a 4 h incubation period with 50 mM tetrazine to induce release. Samples with hEGF only and hEGF–TCO without tetrazine were used as controls. The samples were stored at −20 °C and reaction products were analyzed using a matrix-assisted laser desorption/ionization time-of-flight (MALDI-TOF) mass spectrometer.

The samples were thawed and diluted to 10 and 1 μM of hEGF in H_2_O in a total volume of 5 μL. This mixture was added to an equal volume of a standard α-cyano-4-hydroxycinnamic acid (CHCA) matrix solution (H_2_O:acetonitrile 1:1 + 0.1% formic acid). An initial layer of CHCA solution was applied on the MALDI plate (stainless ground steel 96/12, Bruker Daltonics GmbH & Co., KG, Bremen, Germany), onto which the samples were spotted. The mass spectra of intact proteins were analyzed by MS (Bruker Microflex LRF MALDI-TOF system). The MS data were annotated employing FlexAnalysis software (Bruker).

### 2.4. Ligation of hEGF–TCO to Collagen Scaffolds and Release of hEGF following Tetrazine Exposure

The ability of the hEGF–TCO complex to ligate to an insoluble type I collagen scaffold via the second reactive NHS group was investigated. For each scaffold, 4 μg or 0.8 nmol of hEGF was incubated overnight with 40 nmol TCO in a volume of 20 μL, supplemented with H_2_O. Dry collagen scaffolds, ±1 mg per sample, were wetted overnight in phosphate-buffered saline (PBS, 40 g NaCl, 1 g KCl, 6.9 g Na_2_HPO_4_, 1.3 g KH_2_PO_4_ per L H_2_O, pH 7.2). After wetting of the collagen scaffolds, PBS was removed and replaced with 20 μL of the solution containing hEGF–TCO. Ligation of hEGF–TCO to collagen was performed overnight; afterwards, scaffolds were washed 3 × 10 min in 1 mL PBS while placed on the shaker, after which PBS was removed and replaced with 400 nmol tetrazine diluted in H_2_O to a volume of 20 μL for 4 h. Next, scaffolds were washed 3 × 10 min in PBS and excess fluid was removed. Samples were processed for analysis by sodium dodecyl sulfate polyacrylamide gel electrophoresis (SDS-PAGE) and Western Blotting or immunofluorescence staining as described hereafter.

#### 2.4.1. SDS-PAGE and Western Blotting

Collagen scaffolds were combined with 40 μL 2x reducing sample buffer. Samples were heated for 15 min at 100 °C in a water bath and 10 μL was loaded onto an 8% polyacrylamide gel. The proteins were separated and blotted to nitrocellulose membranes, washed briefly in PBS, and proteins were visualized with 0.1% (*w*/*v*) Ponceau S (Merck, Darmstadt, Germany) in 5% glacial acetic acid in H_2_O. Blots were photographed and destained with PBS with 0.1% Tween20 (PBST, Merck, Darmstadt, Germany) until all Ponceau S was removed. Blots were then blocked in 2% BSA in PBST for 1 h at room temperature under constant agitation and incubated for 1 h with rabbit anti-human EGF (Peprotech, Cranbury, NJ, USA) diluted 1:5000 in blocking buffer. After washing 3 × 5 min in PBS, secondary labeling was performed by incubating blots for 1 h under constant agitation at room temperature with goat-anti-rabbit IRDye 800CW diluted 1:15,000 in blocking buffer (LI-COR Biosciences, Lincoln, NE, USA). Blots were scanned using the Odyssey^®^ CLx infrared imaging system (LI-COR, Lincoln, NE, USA) and accompanying software (Image StudioTM Version 5.0, LI-COR, Lincoln, NE, USA).

#### 2.4.2. Immunofluorescence Staining

Collagen scaffolds were embedded in Tissue-Tek^®^ O.C.T. compound (Sakure Finetek Europe B.V., Alphen aan den Rijn, NL, The Netherlands) and 5 μm cryosections were cut. Sections were blocked for 1 h with blocking buffer containing 2% BSA in PBST. Labeling of hEGF was performed with rabbit anti-human EGF for 1 h (1:200, Peprotech, Cranbury, NJ, USA). Samples were washed 3 × 5 min with PBST and then incubated with Alexa Fluor 488-conjugated goat-anti-rabbit for 1 h (1:200, Molecular Probes Inc, Eugene, OR, USA) in 2% BSA-PBST, washed 3 × 5 min with PBST, and mounted with Mowiol^®^ (Merck). Sections were imaged with a Zeiss Axio Imager A2 microscope (Zeiss group, Oberachen, DE, Germany) and accompanying Zen 2 (blue edition) software.

### 2.5. Ligation of hEGF to BSA and Release following Tetrazine Exposure

Further efforts to visualize the ligation of hEGF–TCO and release of hEGF after tetrazine exposure used BSA as the carrier protein, allowing all reactions to occur in solution. Each reaction contained 3.5 μg or 0.64 nmol hEGF conjugated to 32.14 nmol TCO for 4 h. BSA (7.52 μM) was added to the reaction mixture and allowed to react with the hEGF–TCO complex for 16 h followed by exposure to 321.4 nmol tetrazine for 8 h. Reactions were all carried out in volumes of 20 μL and supplemented with H_2_O where necessary.

The samples were diluted 1:1 in reducing sample buffer and boiled for 15 min to denature proteins. A volume of 10 μL of each sample was loaded onto a 12% acrylamide gel and SDS-PAGE followed by Western blotting as described. After blotting, a Ponceau S stain was applied to visualize total protein load. Blots were washed with PBST until Ponceau S was removed and then blots were blocked in 2% dried skimmed milk (Marvel, Premier Foods, UK) in PBST for 1 h at room temperature under constant agitation. Proteins were labeled for 1 h with rabbit anti-human EGF (1:5000, Peprotech, Cranbury, NJ, USA) and mouse anti-human BSA (1:1000, Sigma Aldrich, St. Louis, MO, USA) diluted in skimmed milk blocking buffer. Visualization was performed by incubating the blots with goat-anti-rabbit IRDye 800CW (1:15,000, LI-COR, Lincoln, NE, USA) and goat-anti-mouse IRDye 680RD (1:15,000, LI-COR, Lincoln, NE, USA) for 1 h in blocking buffer. The blots were scanned using the Odyssey imaging system and software.

## 3. Results

### 3.1. hEGF Ligates to TCO to Form an ‘hEGF–TCO’ Complex

The ability of TCO to couple with hEGF was investigated using MALDI-TOF/MS. The samples were measured in the non-linear mode, eschewing the absolute masses of the molecules, although the differences in *m*/*z* ratios remain accurate. The mass of hEGF in H_2_O was detected at 6248.44 *m*/*z* (Figure 2A), which corresponds to the theoretical mass of 6.22 kDa obtained from hEGF’s amino acid sequence. The sample solvent did not influence the mass spectrum of hEGF as the spectrum obtained in 100% H_2_O was similar to the spectra of hEGF in 50% DMSO/50% H_2_O (Appendix A). Therefore, any changes in the spectra of the reaction samples are due to mass shifts induced by the addition of the TCO molecule.

When hEGF was incubated overnight with TCO, the spectrum returned the mass for free hEGF at 6249.65 *m*/*z* followed by several smaller peaks at increasing *m*/*z* values (Figure 2B). Following the ligation of hEGF to TCO, the predicted mass shift was +308.31 *m*/*z* due to the removal of one NHS group from the TCO molecule (hEGF + C_15_H_18_NO_6_). The peak detected at 6576.51 *m*/*z*, located at hEGF +326.86 *m*/*z*, matches this prediction most closely (hEGF + C_15_H_18_NO_6_ + Na^+^). The other peaks may be the result of a number of TCO adducts. The remaining NHS group on a TCO molecule may be subjected to hydrolysis, leading to a mass shift of +211.23 *m*/*z* (hEGF + C_11_H_15_O_4_). Another possibility may be that one TCO molecule has reacted with its two NHS groups to the same hEGF molecule, which would result in a mass shift of 194.23 *m*/*z* (hEGF + C_11_H_14_O_3_). The presence of three primary amine groups on hEGF could also result in one hEGF molecule carrying multiple TCO molecules. Between hEGF and the peak at 6443.34 *m*/*z* there is a difference of +193.69 *m*/*z*, which corresponds to two bound TCO molecules (hEGF + C_11_H_14_O_3_). Intriguingly, the peak at 6776.47 *m*/*z* may be explained in a number of ways. The mass difference between 6443.34 *m*/*z* and 6776.47 *m*/*z* is +333.13 *m*/*z*, very close to the predicted shift of TCO + Na^+^. On the other hand, between 6576.51 *m*/*z* and 6776.47 *m*/*z*, the difference of +199.97 *m*/*z* matches the corresponding values of two TCO molecules bound to hEGF. Alternatively, the peak at 6776.47 *m*/*z*, hEGF + 526.82 *m*/*z*, may also be the result of hEGF with both an intact TCO and hydrolyzed TCO molecule. Lastly, the peak at 6886.70 *m*/*z* is 310.19 *m*/*z* removed from the peak at 6576.51, a mass difference corresponding to one TCO molecule.

The release of hEGF from TCO was investigated after 4 h of tetrazine exposure (Figure 2C). The main peak of free hEGF was measured at 6253.00 *m*/*z*. Another peak was identified at 6484.73 *m*/*z* (hEGF + 231.73 *m*/*z*), which closely approaches the mass of a hydrolyzed TCO + Na^+^. No other distinctive peaks were present in the spectrum.

In short, when comparing the three mass spectra, it is evident that the addition of TCO to hEGF leads to an increase in distinctive peaks, indicative of the hEGF–TCO ligation products. On the other hand, these peaks diminish in the presence of tetrazine, suggesting the partial release of TCO from hEGF.

### 3.2. The hEGF–TCO Complex Is Able to Ligate to Collagen Scaffolds

After establishing the ability of TCO to couple with hEGF through mass spectrometry, the ability of the second NHS group to bind to collagen was investigated. To achieve this, collagen scaffolds were incubated overnight with hEGF–TCO and thoroughly washed in PBS to remove unbound complexes. The presence of hEGF on collagen was explored through immunofluorescence assays on cryosections and Western blots.

The non-functionalized collagen scaffolds did not stain positive for hEGF, indicating the absence of non-specific binding of the hEGF antibody (Figure 3A). The non-functionalized collagen scaffolds incubated with hEGF revealed that some growth factor bound to collagen without the TCO linker, with the signal being mostly concentrated at the outer edges of the scaffold (Figure 3B). However, when the hEGF–TCO complex was incubated with the collagen scaffolds, a substantial increase in the staining of hEGF on the scaffolds was observed, indicating a hEGF–TCO–collagen conjugation (Figure 3C). Entire scaffold sections showed an even coating of hEGF, with little to no difference between the middle and outer edges of the scaffolds, again illustrating a hEGF–TCO–collagen conjugation. The incubation of the hEGF–TCO–collagen scaffolds with tetrazine did not lead to a noticeable decrease in hEGF signal intensity (data not shown), which could be due to the released growth factor persisting on the collagen, even after its release. Further proof of the ability of hEGF–TCO to ligate to collagen was obtained through SDS-PAGE followed by Western blotting (Figure 3D,E). The total protein load, as visualized through Ponceau S staining (Figure 3D), indicated similar protein loads in the various lanes. The collagen scaffold incubated with only hEGF returned only a weak signal at the bottom of the blot, located in the running front, which marks the presence of free hEGF (Figure 3E). In contrast, the collagen scaffold conjugated with hEGF–TCO showed a very strong signal for hEGF on the collagen bands, as well as a slight signal of free hEGF in the running front. Taken together, these results show that hEGF–TCO indeed binds to collagen. The collagen scaffold incubated with hEGF–TCO and tetrazine displayed a reduction in the hEGF signal when bound to collagen but no clear increase in free hEGF at the bottom of the blot.

### 3.3. Establishing Click-to-Release of hEGF from BSA–TCO–hEGF

In order to investigate the release of hEGF without the interference of non-specific binding to collagen, we conjugated hEGF–TCO to BSA in H_2_O followed by an incubation with tetrazine. This approach allowed all the reactions to occur in solution and enabled equal protein loading across the sample lanes. The samples of hEGF–TCO–BSA with and without tetrazine were separated using SDS-PAGE, after which Western blotting was performed.

On the Ponceau S stain, protein bands were located at the predicted height of BSA at 66.5 kDa and in the running front where free hEGF at 6.2 kDa is located (Figure 4A). After the removal of Ponceau S, the blot was immunolabeled for both BSA and hEGF (Figure 4B–D). The combined channels showed that hEGF is located slightly above the band belonging to BSA. This shift is due to the size difference caused by the ligation of hEGF to BSA (6.22 + 66.5 kDa). When separating the channels (Figure 4C), there is no BSA-specific signal at this protein band, which can be explained by the monoclonality of the BSA antibody, which may not recognize the BSA antigen when its binding site is blocked by the hEGF–TCO complex.

When investigating the hEGF-specific signal (Figure 4D), a reduction in hEGF signal was observed at the top of the blot following addition of tetrazine. An increase in hEGF at the bottom of the blot is difficult to assess due to the oversaturation of the free hEGF signal. The release of hEGF is supported by the Ponceau S-stained blot where a decrease in the total protein load at the top of the blot was observed in combination with an increase in the total protein load at the running front on Ponceau S after tetrazine’s addition (Figure 4A). Together, these stains support the release of hEGF from BSA–TCO–hEGF following exposure to tetrazine. These results also demonstrate that TCO can ligate hEGF not only to type I collagen but also to other proteins such as BSA.

## 4. Discussion

We set out to establish a novel approach for the production of spatiotemporal signaling scaffolds using click-to-release chemistry. The research presented here offers the first proof-of-concept for the feasibility of this approach. The data demonstrate the ability of the bis-NHS-TCO linker to functionalize both type I collagen scaffolds and BSA with a growth factor. Mass spectrometry demonstrated the presence of hEGF–TCO ligation products. In addition, the coupling of the hEGF–TCO complex to a carrier protein was shown. The results indicated the partial “click-to-release” of hEGF after tetrazine exposure.

Although the data are encouraging, we must acknowledge some limitations to this proof-of-concept research. While the data obtained with collagen–TCO–hEGF and BSA–TCO–hEGF clearly indicate that there was a ligation of hEGF via TCO, unexpected ligation products were present in the mass spectra. A more rigorous exploration of the reaction products with mass spectrometry may shed light on the origin of these products. Our data clearly show that both NHS groups are reactive, although the difference in the reactivity of the NHS groups due to the steric hindrance provided by the methyl group on TCO could not be determined. The data indicated a partial ligation between TCO, hEGF, and collagen/BSA. The absence of hEGF dimers on the Western Blot, resulting from the formation of a hEGF–TCO–hEGF complex, suggests that the second NHS group could be compromised. The hydrolysis of NHS groups is a realistic concern in aqueous environments and may be responsible for the low ligation efficiency, which might have impacted the ability to detect click-to-release activity with the methods used. Methods with an increased detection sensitivity compared to immunostainings and Western Blotting will be of benefit in this regard.

Despite the study limitations, this research is a promising first step towards the development of spatiotemporal signaling scaffolds capable of the controlled release of active factors. Previously, researchers have reported the use of bis-NHS-TCO and dimethyl-tetrazine in PBS at 37 °C, reaching 80% release of the cytostatic drug doxorubicin within 30 min [48,49]. Similar release yields may thus be obtained in a biomaterial–TCO system upon the optimization of reaction parameters. The researchers also reported the successful release of the bioactive compound when the reactions were performed in serum and in vivo using a tumor-bearing mouse model. Click chemistry is uniquely suited for in vivo use due to its non-toxic characteristics and its lack of toxicity has been extensively reported. A study by Zhang and colleagues (2016) demonstrated that intravenous injections of high doses of dimethyl tetrazine during a period of 20 days had no effect on the growth of mice [52]. A similar report was published by Rossin and colleagues (2016), which also demonstrated no toxic effects after injections of high doses of tetrazine in mice [49]. The “click-to-release” reaction may be optimized in various ways. TCO molecules are known to be unstable and researchers have attempted to overcome this by reversing the system, i.e., using a tetrazine as the releasing species and TCO as the activator [53]. To this end, they designed a tetrazine-linked antibody–drug conjugate with superior stability and release kinetics, allowing for lower dosages in vivo. A similar approach was used to produce an injectable biopolymer conjugated to tetrazine that reacted with a TCO-linked drug for tumor treatment [54]. Tweaking this system to obtain a bi-functionalized tetrazine molecule to enable selective conjugation to a bioactive molecule on the releasing site and carrier protein on the retaining site could be another way to improve the approach we have described in this paper. Another option could include the use of different tetrazine activators, for which a plethora have already been developed [55]. The reaction rates of TCO–tetrazine pairs are highly influenced by electron-withdrawing groups on the tetrazine molecule. The initial cycloaddition is accelerated by the presence of such electron-withdrawing groups but researchers also found that these groups interfere with the de-caging reaction to release the payload, which they circumvented by designing asymmetric tetrazines [56]. On the other hand, electron-withdrawing groups decrease the stability of tetrazine molecules, which could be mitigated by introducing intra-molecular repulsion [57].

The adaptation and optimization of the TCO/tetrazine click-to-release pair will result in a system with excellent abilities that can be exploited with many different releasing species. With the current bis-NHS-TCO, potential signaling factors with a primary amine group can be easily applied. As such, other bioactive molecules with a primary amine group can be considered as the releasing species. Small molecules, functional analogues of growth factors, have several benefits, such as increased stability, reproducibility, and a lack of immunogenicity [58]. In addition, the in vitro effects of small molecules demonstrated their potential to induce specific cell-signaling effects in the field of skin regeneration [59]. However, NHS esters are also reactive towards secondary amines and carboxylic groups, further increasing the number of bioactive compounds that may be employed [60].

The click-to-release system is not limited to the use of only one signaling factor. Instead, multiple factors that work in tandem may be added to a single biomaterial. Various TCO-linkers and tetrazine activators are available and many companies provide on-demand synthesis of new compounds. The selective release of multiple different bioactive factors from one scaffold can be achieved by employing a combination of TCO–tetrazine pairs. By allowing for the spatial and temporal release of bioactive molecules, including growth factors, the biomaterial will facilitate a simulation of the dynamic presence of effector molecules during skin wound healing.

Applications of this technology beyond skin wound healing are evident. In drug screening, for example, the incorporation of growth factors on scaffolds have improved advanced tumor models [61]. In the field of articular cartilage and muscle tissue engineering, the enhancement of biomaterials with bioactive cues such as growth factors has led to improved cell adhesion and differentiation [62,63]. Applying click-to-release chemistry in other fields may open novel paths towards functionalizing biomaterials and further improving their design.

There are many possibilities to explore in the field of ‘click-to-release’, of which this work only encompasses an initial step. Harnessing the power of ‘click-to-release’ chemistry for the development of smart biomaterials should be given utmost priority and every advancement should be encouraged. Only by continuing experimentation with ‘click-to-release’ and discovering the potential as well as the limitations of this system will a new class of biomaterials come into existence.

## 5. Conclusions

This study shows the use of click-to-release chemistry for the development of regenerative biomaterials by demonstrating the coupling and partial release of human epidermal growth factor (hEGF) bound to either type I collagen scaffolds or bovine serum albumin (BSA) using a bis-N-hydroxysuccinimide-trans-cyclooctene (TCO) linker in combination with dimethyl-tetrazine. Continuing efforts into the development of click-to-release chemistry for biomaterials’ functionalization may benefit many research fields.

## Figures and Tables

**Figure 1 pharmaceutics-14-01991-f001:**
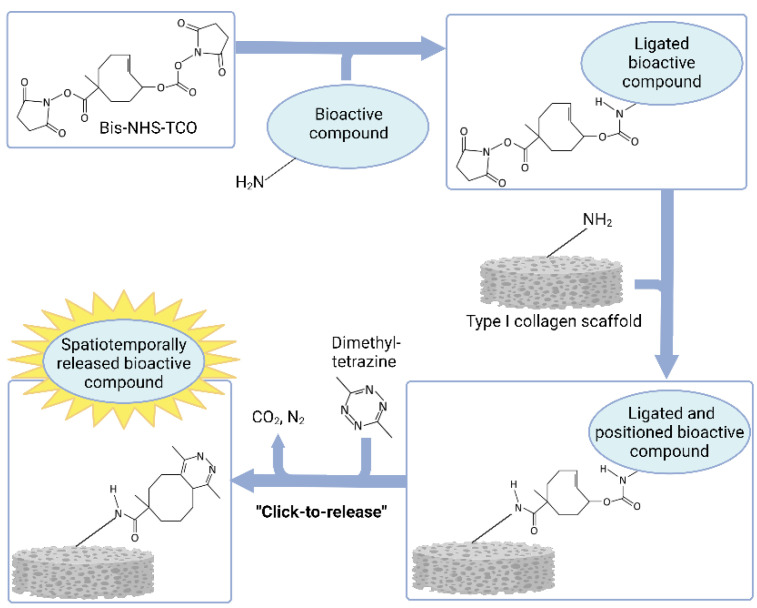
Schematic representation of the functionalization of the bis-N-hydroxysuccinimide (NHS)-functionalized TCO molecule (bis-NHS-TCO) with an amine-containing bioactive compound, followed by conjugation to a collagen scaffold and subsequent spatiotemporal release of the bioactive compound after “clicking” the bis-NHS-TCO with dimethyl-tetrazine. Created with BioRender.com.

**Figure 2 pharmaceutics-14-01991-f002:**
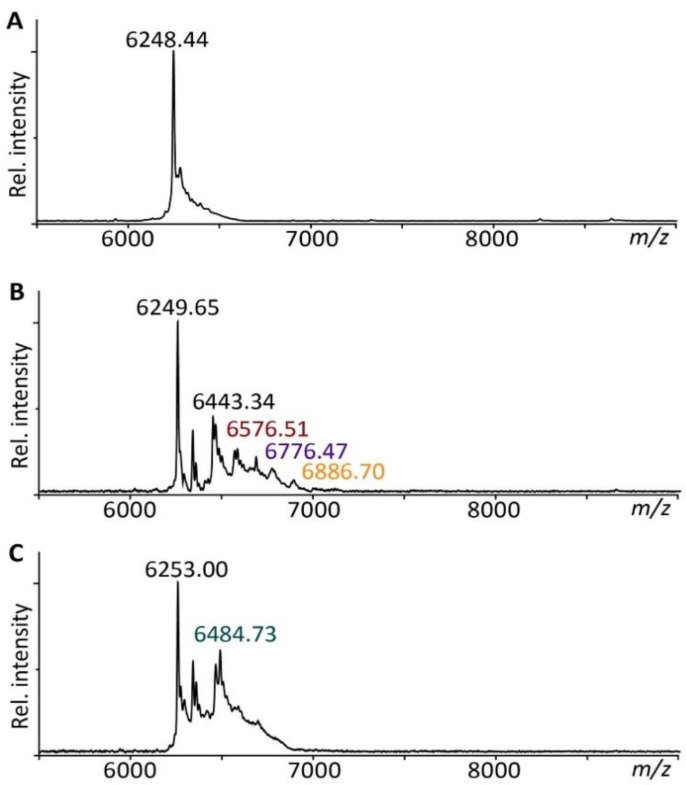
Mass spectra of (**A**) hEGF, (**B**) hEGF and TCO, and (**C**) hEGF–TCO and tetrazine, showing various peaks belonging to free hEGF and EGF with various TCO adducts indicative of successful hEGF–TCO coupling.

**Figure 3 pharmaceutics-14-01991-f003:**
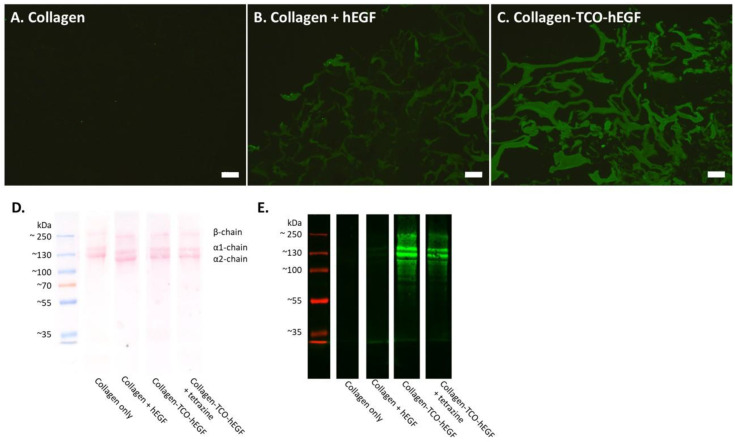
Successful immobilization of hEGF–TCO to collagen scaffolds. Immunofluorescent detection of hEGF on collagen scaffold cryo-sections (**A**–**C**) and Western blotting (**D**,**E**). (**A**) Negative control of collagen scaffold without hEGF, (**B**) collagen scaffold incubated with hEGF only, and (**C**) collagen scaffold conjugated with hEGF–TCO. Scale bar = 50 μm. Green color indicates hEGF. All images were acquired under identical exposure time and laser intensity. (**D**) Total protein load of Western blot of collagen scaffolds is visualized by Ponceau S. (**E**) Western blot of collagen scaffolds, where hEGF is indicated by the green signal. Pre-stained marker stains red under these conditions.

**Figure 4 pharmaceutics-14-01991-f004:**
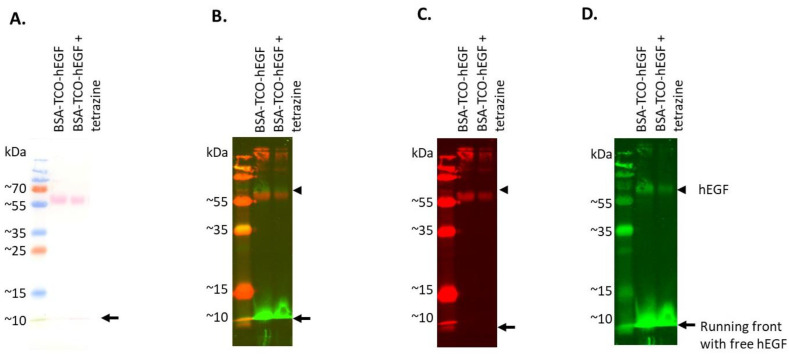
Western blot of samples containing BSA, hEGF–TCO, and tetrazine demonstrate successful hEGF–TCO–BSA coupling and partial hEGF release following tetrazine exposure. (**A**) Total protein load visualized by Ponceau S, (**B**) combined channels of BSA (red) and hEGF (green), (**C**) isolated BSA signal, and (**D**) isolated hEGF signal. Black arrow indicates the running front, the location where free hEGF runs; arrowhead indicates the position of hEGF above the BSA band.

## Data Availability

The data presented in this study are available on request from the corresponding author.

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
