# Peer review of "Initial Steps towards Spatiotemporal Signaling through Biomaterials Using Click-to-Release Chemistry"

_pharmaceutics, 2022, doi:10.3390/pharmaceutics14101991_

Round 1

Reviewer 1 Report

This paper is the technical approach to wound healing or tissue regeneration based on the scaffold and factor release system. This research is interesting and useful for researchers on chemical, tissue engineering, biomaterials, or regenerative medicine. However, the authors should introduce the current technologies or methodologies by utilizing of release system from scaffolds of various representative biomaterials except for collagen. After the section, the authors should mention the strength of this study. In the current version, it is a little difficult to understand the novelty. In addition, this system would be applied to more tissues or diseases. Not only skin application but also many tissues should be mentioned and discussed by quoting the related papers. Taken together, major revision should be made.

1. Introduction

In this field, there are many reports on tissue regeneration based on the release system. The authors should introduce the current technologies with various biomaterials.

I recommend the papers be quoted.

Review papers

Algiante  https://doi.org/10.1002/admi.202100809

Gelatin  Molecules 202126(22), 6795

2.

The authors should investigate the hEGF release profile.

3.

The authors should perform the migration assay or wound healing assay.

4. Discussion

The authors should discuss the feasibility of this technology by comparing some applications not only wound healing.

 I recommend the papers be quoted.

Related review

Cancers    Cancers 202012(10), 2754

Cartilage  https://doi.org/10.1016/j.actbio.2017.11.021

Muscle   https://doi.org/10.1016/j.biomaterials.2019.119584

Reviewer 2 Report

This very well written manuscript describes the application of the so-called "click-to-release" chemistry to the development of innovative pro-regenerative biomaterials for wound healing. In particular, the Authors exploited the fast coupling between the 1,2,4,5-tetrazines and the trans-cyclooctene dienophiles, a very useful click reaction already used for the in vitro and in vivo drug release (see refs. 42-44). The coupling of bis-N-hydroxysuccinimide-trans-cyclooctene (TCO, a molecule already described in the literature) first with (commercially available) recombinant human Epidermal Growth Factor (hEGF) and then with porous collagen scaffolds (prepared as previously reported) was evaluated by MALDI-TOF mass spectrometry and Western blot assays. The same analytical techniques were also employed for the study of the hEGF release upon reaction with tetrazine.

Even if the present approach showed some limitations, the work is interesting and well presented, therefore, it deserves publication in Pharmaceutics without modifications.

Reviewer 3 Report

The manuscript “Initial steps towards spatiotemporal signaling through biomaterials using click-to-release chemistry” refers to the design of a tunable collagen scaffold able to release hEGF by means of click-release chemistry. These scaffolds are intended to be used during wound healing process to promote dermo-epidermal regeneration and avoid scar formation by the action of hEGF. Whereas the chemical description and results of this work are very detailed, and supported by scientific data, there is a lack of biocompatibility/cytotoxicity effect of the scaffolds. As this work is focused in solve the problem of scar formation during wound healing, extra information regarding biocompatibility should be included (at least some preliminary results) using either dermal or epidermal cells to check whether they have really the potential to be used as a wound dressing scaffold. If this work is intended to be used in skin tissue engineering, more specific biological experiments regarding cell behavior should be included. I recommend major revision.

ABSTRACT

In the abstract the authors correctly introduce the limitation of the scaffolds they are testing and the problem they want to solve, as well as the most important results and conclusions. The abstract is very well written and easy to follow.

INTRODUCTION

Very detailed introduction with good references. The authors make a good effort to introduce the problem they want to solve as the same time as provide with the most fundamental information needed to understand the wound healing process and why the choose hEGF molecule.

MATERIALS AND METHODS

Line 185: the reference of the anti-human EGF antibody is not included.

RESULTS AND DISCUSSION

Line 280: the text is between the figure and the legend.

CONCLUSION

Although both the abstract and the introduction of the article focused on wound healing regeneration to justify the coupling of hEGF to collagen matrix by means of click-release chemistry, this is not included in the conclusion. At this point, if the authors want to focus on wound healing, they should provide with more biological data that give preliminary information about the cytotoxic effect onto human dermo-epidermal cells to prove that they can be used with that purpose. If that information is not available, this article may not revolve around the wound healing process and tissue regeneration molecules (hEGF)

FIGURES

Figure 1: low resolution, names of the molecules cannot be clearly seen.

REFERENCES

Enough and good references along the whole article.

Round 2

Reviewer 1 Report

Nice revision version.

Reviewer 3 Report

The authors have succesfully solve all the suggestions as well as answere the question regarding citotoxicity of the compaund. I think that it is a good job, clean and easy to follow.